

# Deep learning with a multi-task convolutional neural network to generate a national-scale 3D soil data product: Particle size distribution of the German agricultural soil-landscape

Mareike Ließ[1,2] and Ali Sakhaee[3]

[1] Department of Agriculture, Food, and Nutrition, Data Science Division, University of Applied Sciences Weihenstephan-Triesdorf, Weidenbach, 91746, Germany
[2] Department Soil System Science, Helmholtz Centre for Environmental Research - UFZ, Halle, 01620, Germany
[3] Thünen Institute of Climate-Smart Agriculture, Braunschweig, 38116, Germany

*Correspondence to*: Mareike Ließ (mareike.liess@hswt.de)

**Abstract.** Many soil functions and processes are controlled by the soil particle size distribution. The generated three-dimensional continuous data product, which covers the particle size fractions of sand, silt, and clay in the agricultural soil-landscape of Germany, has a spatial resolution of 100 m and a depth resolution of 1 cm. This product is an important component for predicting the effects of agricultural management practices and their adaptability to climate change, as well as for analyzing soil functions and numerous risks. The effectiveness of the convolutional neural network (CNN) algorithm in

producing multidimensional, multivariate data products is demonstrated. Even though the potential of this deep learning approach to understand and model the complex soil-landscape relationship is virtually limitless, limitations are data-driven. Further research is needed to assess the required complexity and depth of the CNN and the inclusion of the landscape surrounding each soil profile.

## 1   Introduction

All decisions made in the management of agricultural land and their effects on soil productivity and environmental impact are ultimately influenced by local soil conditions. As a result, information on the soil parameter space at a spatial resolution that addresses individual agricultural fields is needed for the national-scale evaluation and modeling of the influence of agricultural management and climate change on agricultural soils, yields, and the environment (Searle et al., 2021; Ließ, 2022; Dai et al., 2019). This relates to the evaluation of the agricultural productivity of the soils as well as the constraints and necessary

adjustments brought on by protracted drought periods (Mueller et al., 2010). Adequate soil information with high geographic resolution could greatly enhance crop phenology models (Wallach et al., 2022), evaluation and modeling of soil-related drought, and corresponding irrigation requirements (Boeing et al., 2022; Bönecke et al., 2020; Webber et al., 2020). The same is true for assessing the soils' capacity to store soil organic carbon (Chen et al., 2019; Wiesmeier et al., 2019; Sakhaee et al., 2022), modeling the intricate processes that release greenhouse gases to combat climate change (Wang et al., 2021), and



modeling potential solutions to minimize nitrate pollution (Bouraoui and Grizzetti, 2014; Sundermann et al., 2020). The soil
      particle size distribution drives most soil functions and related processes (Vogel et al., 2018; Palm et al., 2007).

      Examples of recent data products providing soil texture information at a national scale are Chaney et al. (2019), Liu et al.
      (2020), Varón-Ramírez et al. (2022), and Schulz et al. (2023). Gebauer et al. (2022) and Ließ (2022) generated soil texture
      data products for the agricultural soil landscape of Germany. The state-of-the-art approach to generating landscape-scale soil

data products involves machine learning to relate soil profile information of a large and well-distributed soil database to gridded
      proxies of the soil forming (SCORPAN) factors climate, organisms, relief, parent material, and time. It is referred to by digital
      soil mapping, predictive soil mapping, or pedometric modeling. Recent reviews are provided by Padarian et al. (2019) and
      Arrouays et al. (2021). However, (Chen et al., 2022) provided a review of digital soil mapping studies at a spatial extent
      >10,000 km$^2$ and found that half of the analyzed articles conducted research on topsoil information only (<30 cm).

To address the aforementioned needs to model and answer questions related to crop growth and quality, impacts of prolonged
      drought, greenhouse gas emissions, and groundwater contamination, three-dimensional (3D) soil data is required. Soils are
      intricate systems that vary in three dimensions. They are identified by a vertical division into soil horizons. The number, size,
      and characteristics of these horizons differ across the landscape. Due to numerous physical, chemical, and biological processes,
      many of their features are also interconnected both inside and between each horizon. The simultaneous modeling of these soil

properties and their changes in horizontal space and depth would be best suited to address this complexity, as it would allow
      for the consideration of target dependencies (Waegeman et al., 2019; Ließ, 2022). On the contrary, the separate modeling of
      related properties may lead to inconsistencies. Particle size distribution itself is not a single value target variable but a
      composition of the weight proportions of different particle size fractions. Among other methods, their simultaneous modeling
      has been addressed using Bayesian maximum entropy (D'Or et al., 2001).

The following succinctly describes various methods for modeling the three-dimensional soil parameter space: The '2.5D
      approach' involves creating separate models for distinct soil properties at predefined soil depths and then combining the spatial
      forecasts. Liu et al. (2020), Varón-Ramírez et al. (2022), and Taghizadeh-Mehrjardi et al. (2020) provide recent examples. The
      'depth function approach' fits a continuous mathematical function over the available horizon data before projecting the
      parameters of the function into space (Bishop et al., 1999). '3D regression kriging' simulates the spatial trend and spatial

autocorrelation (Poggio and Gimona, 2017). Overall, convolutional neural networks (CNN) are becoming more and more
      common for multi-target machine learning, extending both the depth function technique and the 2.5D approach (Behrens et
      al., 2018; Padarian et al., 2018; Wadoux, 2019). However, their application for multi-target prediction in the related field of
      soil spectroscopy is much more common. Initially, CNNs were developed and are widely applied in the fields of image analysis
      and computer vision. Please refer to Aloysius and Geetha (2018), Zhang et al. (2016), and Sakib et al. (2018) for a brief history

and reviews. Applied to soil mapping the approach was adapted to consider the wider landscape context of the respective soil
      profiles to establish models for continuous prediction (Wadoux et al., 2019; Behrens et al., 2010; Taghizadeh-Mehrjardi et al.,
      2020; Behrens et al., 2014). A landscape section in terms of a squared window surrounding each profile site is clipped from
      the data cube of gridded predictor data. These patches are analogous to the input images for a CNN image classification task.



Therefore, the contextual landscape characteristics at the soil profile locations including spatial autocorrelation in the predictor
space are used to derive the soil-landscape relationship via a CNN. Like many other machine learning algorithms, CNNs can
only develop their full potential by applying an optimization approach for hyperparameter tuning (Gebauer et al., 2022). Still,
few researchers have attempted to tune the CNN hyperparameters in predictive soil mapping studies, despite recent work
showing the importance  (e.g. Wadoux et al., 2019; Omondiagbe et al., 2023; Taghizadeh-Mehrjardi et al., 2020).

We will demonstrate an approach for implementing multivariate regression to generate a national-scale data product of the 3D
spatial variation of the particle size distribution for the agricultural soil-landscape of Germany. It will be obtained with a single
model employing a patch-wise multi-target CNN to predict three particle size fractions, sand, silt, and clay simultaneously at
high vertical resolution until 100 cm depth. Genetic algorithm optimization will be applied for hyperparameter tuning. A CNN
model to generate a topsoil data product (2D) is included to provide a benchmark since often the more complex 3D model
training results in a lower performance.

## 2     Material and methods

### 2.1     Landscape setting

In the 357.6 km$^2$ that make up Germany, 51% of the land is being used for agriculture (Federal Statistical Office of Germany,
2022). From north to south, its landscape is divided into four morphologic areas: the North German Lowland, the Central
German Uplands, the Alpine Foreland, and the Alps. The majority of the North German Lowland is located below 100 m a.s.l.
The North-Eastern region of Germany has glacial influence with many lakes and moraines, in contrast to the marshes,
peatlands, and marshy terrain around the North Sea Coast. The central German Uplands region's mountains are generally not
much higher than 1,000 meters above sea level. They developed basin structures with sedimentary deposits under the influence
of several periods of upheaval and subsidence and are interspersed with alluvial glacial loess deposits. In addition, many of
the mountain ranges show evidence of past volcanism, creating a complex geological structure. The Alpine Foothills region,
with elevations between 400 and 750 meters above sea level, was shaped by glaciers and exhibits a wide range of
geomorphological features, including molasses basins with sedimentary deposits from Alpine erosion and morainic hills and
aprons. The Northern Calcareous Alps include the portion of the Alps in Germany. Parent material and topography are the
main determinants of soil distribution in Germany. Ließ et al. (2021) and Gebauer et al. (2022) provide further details on the
German landscape setting.

### 2.2     Soil data

Data on the particle size distribution of 3102 soil profiles were available from the German agricultural soil inventory (First
German Agricultural Soil Inventory – Core dataset; Jacobs et al., 2018). The data was collected nation-wide through systematic
sampling along an 8 by 8 km grid. Samples were taken for each of the 0-10, 10-30, 30-50, 50-70, and 70-100 cm depth
increments, while taking horizon boundaries into account. The particle size distribution in terms of three particle size fractions



– sand (2,000-63 μm), silt (63-2 μm), clay (<2 μm) – was determined by sieving and the sedimentation method according to DIN ISO 11277 (2002). The data of the three particle size fractions was subdivided into 1 cm slices and included as response data in model training, tuning and evaluation.

## 2.3 Data to approximate the soil forming factors

The covariates used to train the machine learning models for nationwide spatial prediction were grouped according to the
SCORPAN factor they represent. Table 1 gives an overview including references.

The German Weather Service provided seasonal averages of air temperature and drought, as well as the sum of precipitation for the winter (Dec., Jan., and Feb.) and summer (Jun., Jul., and Aug.) months for SCORPAN C.

The following factors were included to approximate SCORPAN O: Sentinel-2 data composites from the second yearly quartiles of 2018 and 2021 of the bands B01, B02, B03, B04, B05, B06, B07, B08, B8a, B11, and B12, as well as the vegetation indices
Enhanced Vegetation Index (EVI), Moisture Index (MSI), Normalized Difference Moisture Index (NDMI), Normalized difference vegetation index (NDVI), Normalized difference water index (NDWI), and Plant Senescence Reflectance Index (PSRI) (details in Table 1). The composites were created using the Sentinel-Hub and the surface reflectance values from the Level 2A product. Before computing the vegetation indices, the composites were downloaded as multiple tiles with a spatial resolution of 20 m, then mosaicked, and resampled to the 100 m INSPIRE — Infrastructure for Spatial Information in Europe
— grid topology (INSPIRE TWG, 2014). Furthermore, Copernicus Global Land Service remote sensing products on dry matter productivity (DMP) and the Vegetation Productivity Index (VPI) during the time period June 11th-20th, 2016 and 2018 were obtained. All the SCORPAN O variables were selected to reflect the major annual phase of agricultural productivity.

SCORPAN R was represented by the Geomorphographic map of Germany and terrain parameters produced from the EU-DEM digital elevation model using digital terrain analysis with the SAGA — System for Automated Geoscientific Analyses
(Conrad et al., 2015).

To approximate SCORPAN P, the map of "Groups of soil parent material" was incorporated. The hydrogeological map of Germany's lithology and stratigraphy were also added.

Proxies to soil (SCORPAN S) can be found in conventional soil polygon maps and remote sensing products related to soil parameters. In the case of the former, a map of German soilscapes was included. Differences in DMP and VPI between the
dry year 2018 and the fairly wet year 2016 were incorporated in the latter. They are related to crop phenology and, as a result, to the root zone plant-available soil water capacity.

All covariates were resampled at 100 m resolution to the INSPIRE grid topology (INSPIRE TWG, 2014). This resolution was chosen as a compromise between the desire to provide soil information for individual agricultural fields and the usage of computational resources in a limited manner. For categorical predictors, the nearest-neighbor approach was utilized, while for
numerical predictors, B-spline interpolation was used. INSPIRE latitude and longitude were also supplied to reflect the geographic location (SCORPAN N), and in particular to represent spatial patterns that the other data proxies did not capture.



The German national boundary and shoreline were obtained using the Federal Agency for Cartography and Geodesy's digital land model at map scale of 1:250,000 (version 2.0) (GeoBasis-DE / BKG, 2020).




**Table 1: Covariates**

| Soil forming factor | Abbreviation | Description | Data source |
|---|---|---|---|
| Climate | PRESU | Average seasonal precipitation (summer) [raster, 1000 m] | DWD (2018b) |
| | PREWI | Average seasonal precipitation (winter) [raster, 1000 m] | |
| | TEMSU | Average seasonal temperature (summer) [raster, 1000 m] | DWD (2018a) |
| | TEMWI | Average seasonal temperature (winter) [raster, 1000 m] | |
| | DINSU | Average seasonal drought index (summer) [raster, 1000 m] | DWD (2018c) |
| | DINWI | Average seasonal drought index (winter) [raster, 1000 m] | |
| Organisms/ Soil | B0118, 0218, …B0818, B8A18, B1118, B1218 | Sentinel-2 spectral bands B1, B2,…B8, B8A, B11, and B12 composites of the 2nd yearly quartile of the year 2018 | https://www.europeandataportal.eu |
| | B0121, 0221, …B0821, B8A21, B1121, B1221 | Sentinel-2 spectral bands B1, B2,…B8, B8A, B11, and B12 composites of the 2nd yearly quartile of the year 2021 | https://www.europeandataportal.eu |
| | EVI18, EVI21 | Enhanced vegetation index, calculated from Sentinel 2 band composites of 2nd quartile 2018 & 2021 (S2-Q2-18/21), $EVI = G*(B8A-B04)/(B8A + C1*B04 - C2*B02 +L)$, with $G = 2.5$, $C1 = 6$, $C2 = 7.5$ and $L = 1$ | |
| | MSI18, MSI21 | Moisture index: S2-Q2-18/21, $MSI = B11/B08$ | |
| | NDM18, NDM21 | Normalized difference moisture index: S2-Q2-18/21, $NDMI = (B08-B11)/(B08+B11)$ | |
| | NDV18, NDV21 | Normalized difference vegetation index: S2-Q2-18/21, $NDVI = (B08-B04)/(B08+B04)$ | |
| | NDW18, NDW21 | Normalized difference water index: S2-Q2-18/21, $NDWI = (B03-B08)/(B03+B08)$ | |
| | PSR18, PSR21 | Plant senescence reflectance index: S2-Q2-18/21, $PSRI = (B04-B02)/B06$ | |
| | DMP16 | Dry matter productivity, June 2016 [raster, 300 m] | Swinnen and Van Hoolst (2019) |
| | DMP18 | Dry matter productivity, June 2018 [raster, 300] | |
| | VPI16 | Vegetation Productivity Index, June 2016 [raster, 300 m] | Swinnen and Toté (2015) |
| | VPI18 | Vegetation Productivity Index, June 2018 [raster, 300 m] | |

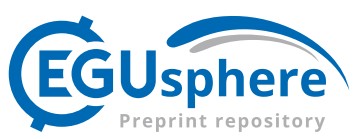

**Table 1 (continued): Covariates**

| Soil forming factor | Abbreviation | Description | Data source |
|---|---|---|---|
| Topography | GMK00 | Geomorphographic map of Germany [raster, 250 m resolution, map scale 1:1,000,000] | (BGR, 2007) |
| | DEM00 | Digital elevation model [raster, 25 m resolution] | (European Environment Agency (EEA), 2017) |
| | SLO01, SLO05, SLO10 | Slope: calculated from DEM (cfD) with a search radius of 1, 5, 10 cells, using SAGA module Morphometric features | |
| | NOR01, NOR05, NOR10 | Northness — derived from aspect cfD with a search radius of 1, 5, 10 cells, using SAGA module Morphometric features | |
| | EAS01, EAS05, EAS10 | Eastness | |
| | TST01, TST05, TST10 | Terrain surface texture: cfD with a search radius of 1, 5, 10 cells, using SAGA module Terrain Surface Texture | |
| | TSR01, TSR05, TSR10 | Terrain surface ruggedness: cfD with a search radius of 1, 5, 10 cells, using SAGA module Terrain Ruggedness Index | |
| | CON01, CON05, CON10 | Convergence Index: cfD with a search radius of 1, 5, 10 cells, using SAGA module Convergence Index (Search Radius) | |
| | SLH00 | Slope Height — cfD using SAGA module Relative Heights and Slope Positions | |
| | VAD00 | Valley depth | |
| | NOH00 | Normalised Height | |
| | WIN00 | Wind Exposure: cfD using SAGA module Wind Effect | |
| | NOP00 | Negative openness — cfD using SAGA module Topographic Openness | |
| | POP00 | Positive openness | |
| | VOF0S | Vertical overland flow distance to all river segments — cfD using SAGA module Terrain analysis / Channels | |
| | VOF0M | Vertical overland flow distance to major rivers | |
| | HOF0S | Horizontal overland flow distance to all river segments | |
| | HOFOM | Horizontal overland flow distance to major rivers | |
| | SW100 | SAGA wetness index: cfD using SAGA module SAGA Wetness Index | |
| Parent material | LIT00 | Lithology, Hydrogeological map of Germany, HÜK [polygon shapefile, map scale 1:250,000] | (BGR and SDG, 2019) |
| | STR00 | Stratigraphy, Hydrogeological map of Germany, HÜK [polygon shapefile, map scale 1:250,000] | |
| Soil | BAG00 | Groups of soil parent material in Germany [polygon shapefile, map scale 1:5,000,000] | (BGR, 2008a) |
| | BGL00 | Soil scapes in Germany [map scale 1:5,000,000] | (BGR, |
| | DMP86 | Dry matter productivity, DMP18–DMP16 [raster, 300 m] | |
| | VPI86 | Vegetation Productivity Index, VPI18–VPI16 [raster, 300 m] | |
| Geographic location | LAT00 | INSPIRE Latitude | (INSPIRE TWG, 2014) |
| | LON00 | INSPIRE Longitude | |





## 2.4 Convolutional neural network

CNNs are feed-forward artificial neural networks (ANNs) trained with back-propagation (Le Cun et al., 1989, 1998). ANNs
consist of an input layer, an output layer, and one or more hidden layers. The learning process builds up the network structure
to generalize the relation between the input (the covariate information at the soil profile sites) and the output (the soil texture
at the profile sites). The hidden layers consist of a certain number of neurons. A neuron is the basic unit of an ANN. It is
connected to all neurons of the previous and all neurons of the subsequent layer. Weights are assigned to the respective
connections to give certain inputs a higher importance compared to others. In addition, a bias is assigned to each neuron. An
optimization technique, often referred to as an optimizer, is applied to update the weights during the learning process in order
to achieve the lowest prediction error. Finally, the learning rate determines how fast the weights are updated. A high value
speeds up the learning process, while a low value makes sure the network succeeds in learning the predictor-response relation.
The application of an activation function at each neuron transforms the input into the output. It allows for the introduction of
non-linearity. A dropout rate is commonly applied for the regularization of the network structure. This means that a certain
portion of randomly selected neurons are ignored during training. Accordingly, their contribution is temporarily removed, and
weight updates are not applied. Dropout reduces co-adaptations of neurons.

CNNs allow for the inclusion of different types of input data. Instead of the $n$ x $p$ covariate matrix representing $p$ covariate
values extracted at the $n$ profile sites, CNNs allow for the inclusion of covariate data arrays $X$ with dimension [n, w, w, p].
Each profile site is represented by an array [1, w, w, p], where w is the window size defining the surrounding landscape patch.
CNNs consider the information contained in the $w$ x $w$ landscape surrounding in terms of the spatial autocorrelation between
the contained raster cell values and their spatial structures to detect higher-order features (Le Cun et al., 1989). Accordingly,
the hidden layers in a CNN include layers that perform convolutions. As the convolution kernel slides along the input data for
the layer, the convolution operation generates a feature map, which in turn contributes to the input of the next layer. Each unit
in a feature map takes inputs from a subregion, i.e. receptive field, of the original input data, acting as feature extractors. The
input values from the receptive field are linearly combined using the weights and the bias and the result is transformed by the
activation function. All units of a feature map share the same weights. With a subregion size of e.g. 3 x 3, a feature map has 9
adjustable weight parameters and one bias parameter. A filter is made up of these parameters. Because of the weight sharing,
evaluating the activation function of these units is equal to combining the cell values with a kernel made up of the weight
parameters (Bishop, 2006). Usually, a convolutional layer consists of multiple feature maps, each with its own set of weight
and bias parameters. A pooling layer then follows the convolutional layer. Each feature map of the convolutional layer
corresponds to a plane in the pooling layer. The information in each receptive field is often pooled by extracting its maximum,
i.e. MaxPooling. Then, the output is flattened before it enters a sequence of dense layers (the layers in an ANN).

Figure 1 displays the CNN structure which was used to train the models to predict soil texture in 2D (topsoil) and 3D. All
processing was done in R 4.1.3 (R CoreTeam, 2022), using the 'keras' package (Kalinowski et al., 2022) to connect to the
Keras Python API (Chollet, F., 2015). Patch sizes of $w$ = 5 x 5 and 10 x 10 cells were tested to incorporate the surrounding



landscape context. It is a compromise as the consideration of the surrounding area for each profile site leads to missing values for those sites close to the border of Germany. CNNs cannot handle this type of input. Therefore, this leads to a reduction in the considered profile sites and a reduction in the predicted area. MaxPooling was applied with a receptive field of 2 x 2. The rectified linear unit (ReLU) activation function was applied and the Adam optimizer (Kingma and Ba, 2015) was used to

update the weights during the learning process to achieve a minimum mean squared error. One hundred epochs were used for training. Due to the prediction of the particle size distribution with three particle size separates in 2D (topsoil) and 3D (continuously from 0 to 100 cm with a resolution of 1 cm), two different data structures for the response Y were used: $Y$ [n, 3] for 2D and $Y$ [n, 100, 3] for 3D. The Softmax activation function was employed at the end of the final layer. It serves to map the predictions to non-negative values and to fulfill the constraint that the contributions of all particle size fractions need to

sum up to 1 (Aloysius and Geetha, 2018). Hyperparameter tuning by genetic algorithm optimization was conducted to select the number of filters ($p_1$), the kernel size ($p_2$), the dropout rate ($p_3$), the number of units in the first and second subsequent fully connected dense layer (p4, p5) and the learning rate (p6). Table 2 provides the hyperparameter ranges.

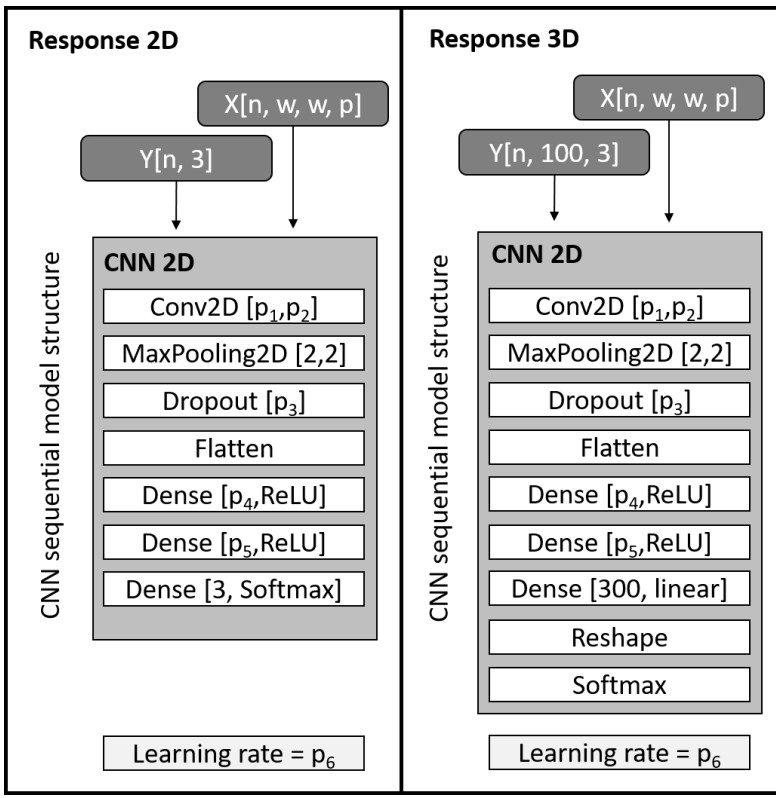

**Figure 1: CNN input data (X – covariates, Y – response), model structure, and hyperparameters (p1, p2, p3,… p6).**




**Table 2: Ranges for hyperparameter tuning**

|    | Hyperparameter | Minimum | Maximum |
|----|----------------|---------|---------|
| P1 | Number of filters | 25 | 100 |
| P2 | Kernel size in both dimensions | 2 | 4 |
| P3 | Dropout rate | 0.1 | 0.5 |
| P4 | Number of units | 25 | 100 |
| P5 | Number of units | 25 | 100 |
| P6 | Learning rate | 0.001 | 0.01 |

## 2.5    Model training, tuning, and evaluation

Table 3 indicates the size of the datasets used for model training to predict soil texture in 2D and 3D while considering a smaller and larger landscape patch surrounding each soil profile. The different sizes result from excluding predictor arrays with missing data, and/ or profiles with missing texture data in part of the profile. The latter results from organic soils or mineral soils with organic horizons, while the former mainly correspond to profile sites close to the national border or the sea,

with a distance of <250 m (w=5) and <500 m (w=10). In general, SCORPAN data proxies are often limited to national borders and not available beyond, which hampers data science approaches depending on these data. Accordingly, the dataset for 2D [5 cells] model development is the largest, while the dataset for 3D [10 cells] is the smallest. The decrease from 2917 to 2740 sites comparing the datasets for 2D and 3D predictions [5 cells] is due to missing texture data in part of the subsoil. The decrease from 2917 to 2679 [2D] comparing the datasets with 5 and 10-cell patch sizes is due to missing data in the predictor

arrays.

**Table 3: Size of the predictor arrays**

| Predictions | w | Predictor array size | Response array size |
|-------------|---|----------------------|---------------------|
| 2D | 5 cells | [1:2917, 1:5,   1:5,   1:119] | [1:2917,1:3] |
| 2D | 10 cells | [1:2679, 1:10, 1:10, 1:119] | [1:2679,1:3] |
| 3D | 5 cells | [1:2740, 1:5,   1:5,   1:119] | [1:2740,1:100,1:3] |
| 3D | 10 cells | [1:2510, 1:10, 1:10, 1:119] | [1:2510,1:100,1:3] |

Model training and evaluation were conducted by a 5-times repeated stratified 5-fold cross-validation (CV) to obtain robust models (Hastie et al., 2009). To implement model tuning, the CV was nested. The predictor-response dataset was subdivided into five folds of equal size. Of these five folds, one fold was always kept as a test set while the other four were combined to

form the model training set, leading to five separate test set evaluations (one per data instance). Each of the outer CV's training



sets was again further subdivided to provide the datasets for hyperparameter tuning in the inner CV cycle. To save time and computational resources, only the first of the 5 repetitions of the inner CV cycle was used for optimizing the hyperparameters. To avoid autocorrelation effects between training and test set data, the data was split considering the profile as an entity. This ensures that a sample from an upper horizon will not be used in the validation when evaluating the precision of a lower horizon

of that profile and vice versa.

Concerning the categorical predictors, all predictors were recoded into dummy variables. Categories not present in all data subsets were removed before model training, tuning, and evaluation. To prevent numerical issues and imbalance, all numerical data were scaled to the range 0–1 using min-max scaling. Table 3 indicates that altogether 119 predictor variables remained. The response data were stratified by partitioning around medoids clustering (Kaufman and Rousseeuw, 1990) with R package

'cluster' (Maechler et al., 2022). The Silhouette Index (Rousseeuw, 1987) was used to determine the number of clusters ≥4 and ≤10.

## 2.6    Genetic algorithm optimization

Hyperparameter tuning was carried out using a genetic algorithm (GA). The operational structure of GAs is inspired by general biological evolution principles such as mutation, crossover, selection, and elitism (Affenzeller et al., 2009).

The parameter space that is searched for the best combination of hyperparameter values must be set by specifying a minimum and maximum value for each parameter. The parent population is then evaluated by a problem-specific objective function using a random number of n parameter vectors. Weights are applied to each parameter vector based on its objective function value before beginning to modify them through selection, mutation, and crossover to create a new population of parameter vectors that is evaluated again. This method is repeated until either (1) any of the vectors attain an initially defined objective

function value, (2) a maximum number of iterations is reached, or (3) the overall best objective function value does not improve for a certain amount of iterations. GA optimization was performed in parallel, subdividing the parent population of 500 individuals into subpopulations and allowing for restricted exchange of population members (hyperparameter vectors) between the islands.

A few test runs were performed to select the GA parameters based on Scrucca (2017) recommendations. The population size

was set to 500 parameter vectors, and the number of islands for simultaneous search was limited to 25. The migration interval between islands was set to 18 and the migration rate to 0.1. With a probability of 0.8, single-point crossover between parameter vectors was performed, and uniform random mutation with a frequency of 0.1. Elitism allowed the best five individuals to survive at each generation. The likelihood of using local search was set to 0.1, and the selection pressure was set to 0.7. The total number of iterations was set to 200, and the number of consecutive generations without an improvement in the best fitness

value before the GA was terminated was set to 20. The average squared prediction error over all targets as a loss function was minimized. The procedure was carried out with the help of the R package 'GA' (Scrucca, 2017, 2019).




## 2.7     Variable importance

Each predictor's importance was determined by permuting the predictor in the test set prior to model application. Any predictor-response association relating to that predictor was thus deleted. The resulting relative increase in the predictive RMSE was

then assigned to the respective predictor as variable importance (VI). This VI estimation was performed for each of the three particle size fractions as well as for all depth slices (3D prediction). The values from five permutations were averaged. The VI values for the dummy variables were generated by aggregating each categorical predictor. The VI plots exhibit boxplots of twenty-five VI values for each predictor due to the five times repeated 5-fold CV procedure (outer CV cycle).

## 3     Results and discussion

## 3.1     Soil data

Figure 2 indicates a high variability of the soil texture throughout the sampled soils in Germany. Particularly sand contents vary between 0 and 100%. The variability is much lower for the clay content. Overall, the soils under agricultural use mostly have a clay content below 27% as indicated by the maximum of the upper hinge in Fig. 2C. Regarding the 2740 profiles included in the dataset for 3D modeling with a 5 × 5 cell patch size, field data annotations show that 44.8% of the soil profiles

have one change in parent material up to a depth of 100 cm, and an additional 14.1% have two or more changes. Only 1.8% of the profiles have the first change in parent material at ≤25 cm depth, 25.1% have it between 25 and 50 cm, 25.5% have it between 50 and 75 cm, and 47.5% have it below 75 cm.

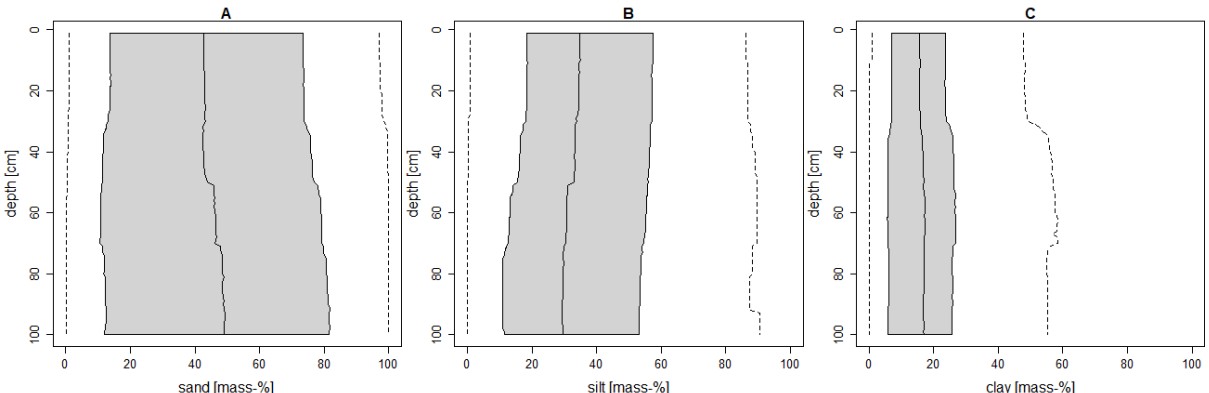

**Figure 2: Distribution of the particle size fractions in the soil profile dataset (2740 profiles). The solid line indicates the median of**
**the distribution, the shaded area between dotted lines reflects the area between the upper and lower hinge, and the dotted lines correspond to the upper and lower whisker. A) sand content, B) silt content, and C) clay content.**

## 3.2     Model hyperparameters

Table 3 provides the results of model hyperparameter tuning with GA optimization. The Learning rate and kernel size resulted in the overall lowest value for model training with a 2D and 3D response for both patch sizes, w = 5 and w = 10 cells, with P2



= 2, and P6 = 0.001. The same applies to the dropout rate (P3 = 0.1) for the topsoil models (2D). For the models with the 3D response, the value is slightly higher but still low. The selected number of filters was between 87 and 91 and was slightly higher for the models considering w = 5 cells patch size compared to those with w = 10 cells patch size. The same applies to the number of units with the exception of the 3D [10 cells] model.

**Table 3: Selected model hyperparameters**

|  | Parameter | 2D [5 cells] | 2D [10 cells] | 3D [5 cells] | 3D [10 cells] |
|---|---|---|---|---|---|
| P1 | Number of filters | 91 | 87 | 93 | 88 |
| P2 | Kernel size in both dimensions | 2 | 2 | 2 | 2 |
| P3 | Dropout rate | 0.1 | 0.1 | 0.1019826 | 0.1606008 |
| P4 | Number of units | 75 | 59 | 71 | 86 |
| P5 | Number of units | 75 | 60 | 52 | 46 |
| P6 | Learning rate | 0.001 | 0.001 | 0.001 | 0.001 |

Wadoux et al. (2019) and Taghizadeh-Mehrjardi et al. (2020) used a more complex CNN structure with more convolutional layers. Wadoux et al. (2019) used two, and Taghizadeh-Mehrjardi et al. (2020) used even four convolutional layers. We refrained from doing so due to the decision to restrict the maximum patch size for previously mentioned reasons. The high number of selected filters could possibly indicate that the models would benefit from an additional convolutional layer. The first convolutional layer's kernels are built to recognize basic characteristics like edges and curves, whereas the kernels in

subsequent layers are trained to recognize more complex features (Zhang et al., 2019). We may gradually extract higher-level information by stacking many convolutional and pooling layers. The importance of additional convolutional layers and hence the necessity of a more complex model structure would have to be tested, though, in a setting where predictor information is available beyond the boundary of the research area and an increase in patch size would not reduce the number of profile sites. Taghizadeh-Mehrjardi et al. (2020) tested patch sizes of 3 to 29 cells and received the best results for a patch size of 7 cells.

Wadoux et al. (2019) tested selected patch sizes from 3 to 35 cells and achieved the best results with a patch size of 21 cells. However, this likely depends on the respective landscape and available predictor information. Patch size and predictor resolution likely affect each other.

### 3.3    Predictive performance

Figure 3 displays the uncertainty of the model predictions. The overall model performance of the topsoil predictions, within

the top 30 cm, (Fig. 3A, 3B, and 3C) is better when considering a smaller landscape surrounding for each profile site. The median RMSE is 17.2 mass-% for the sand, 14.0 mass-% for the silt, and 9.5 mass-% for the clay content. For the 3D predictions, the result is different. For the sand and clay content, the larger landscape surrounding of the 10 cells' patch provides the lower test set RMSE values throughout depth. The median RMSE is 17.8 mass-% for the sand, 14.4 mass-% for the silt,





and 9.3 mass-% for the clay content up to a depth of 10 cm. The average median RMSEs for the top 30 cm of this model are

18.0, 14.5, and 9.5 mass-% respectively.

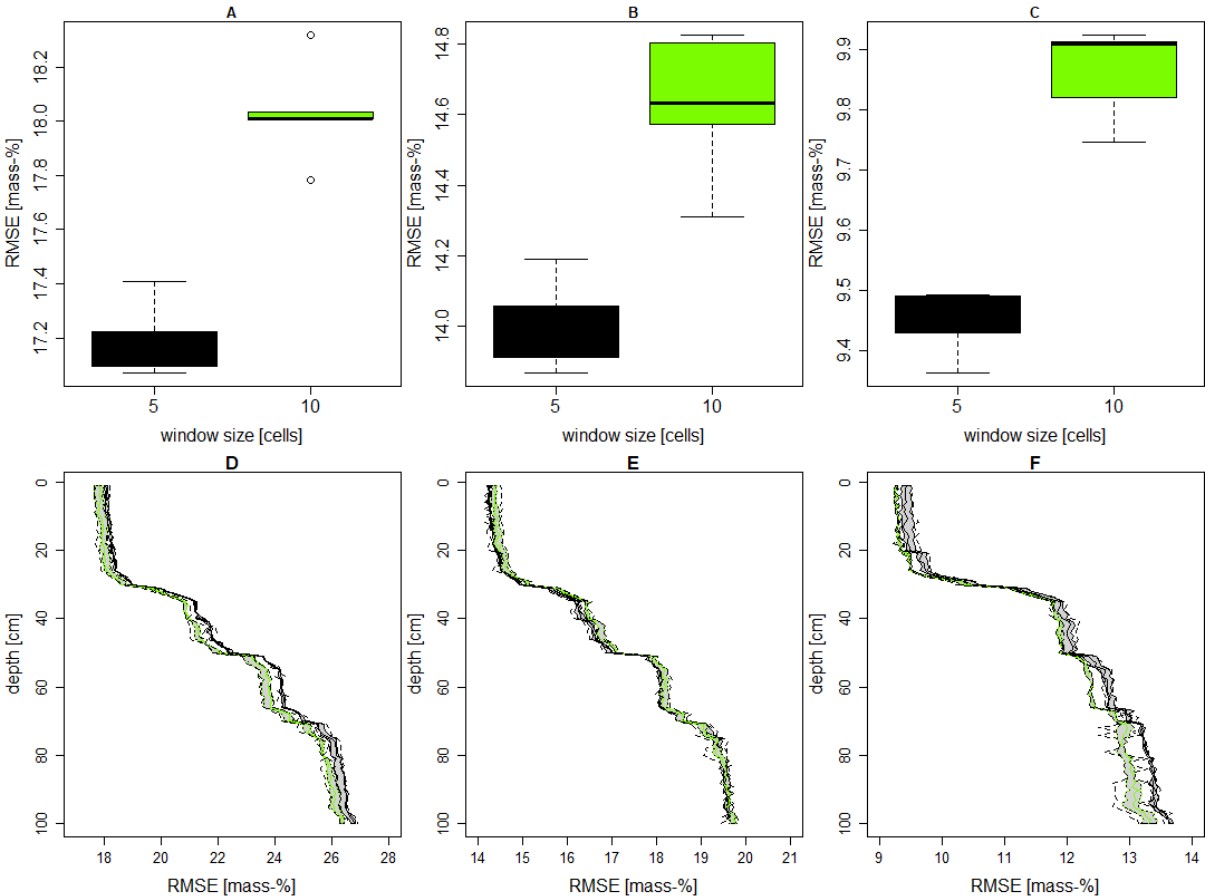

**Figure 3: Predictive model performance. The upper line of figures corresponds to the 2D texture predictions: A = sand content, B = silt content, C = clay content. The lower line of figures corresponds to the 3D texture predictions: D = sand content, E = silt content, F = clay content. The colors reflect the two patch sizes. Black w = 5 cells, and green w = 10 cells.**

Below 26 cm, the RMSE increases abruptly. At 40 cm depth, the RMSE values are 20.9, 16.5, and 11.8 mass-%. The decrease

in model performance with depth has often been reported (Taghizadeh-Mehrjardi et al., 2020; Ließ, 2022; Poggio and Gimona,

2017). One reason could be the frequent changes in parent material. Considering the high percentage of profile sites with

changes in parent material it is likely that the available data for SCORPAN P is causing this increase in uncertainty. The

SCORPAN P proxies consist of geological maps of different scale and quality, and there may be some inconsistencies in how

depth is represented. These maps may not fully capture the nuances of the underlying parent material. Still, it could possibly

be replaced by topographic information to some extent as was shown in various studies. Topographical predictors derived from

a DEM have the advantage that they contain measured data in contrast to the expert information on parent material.



Gebauer et al. (2022) achieved a slightly better model performance for the 30 cm topsoil predictions of the agricultural German soil-landscape on behalf of the same dataset with 15.0 mass-% (sand), 11.8 mass-% (silt), and 8.2 mass-% (clay). An important

factor leading to the comparatively lower model performance in our case could be the reduced size of the soil dataset, which was a consequence of the consideration of the landscape information surrounding each sampling site. This impact was even stronger when increasing the patch size. We do not see another reason since both studies rely on the data from the German agricultural soil inventory, use powerful algorithms, and apply optimization for parameter tuning.

The SoilGrids 2.0 data product (Poggio et al., 2021) has an uncertainty of 19.3 – 25.9 mass-% (sand), 16.5 – 19.6 mass-%

(silt), 11.4 – 14.3 mass-% (clay) increasing from the top layer of 0 – 5 cm to the bottom layer of 60 – 100 cm (Ließ, 2022) with a slightly higher uncertainty. Global scale models cannot include the same level of detail in the predictor information and often do not have access to the same amount of soil profile data as national approaches. And the result of any data science approach essentially depends on the considered data, besides the chosen modeling approach.

Unfortunately, the boundary restrictions on the predictor data did not allow us to test larger patch sizes which could possibly

have further enhanced predictive performance. Wadoux et al. (2019) handled missing data entries by assigning them a value of −1. However, we find it difficult to justify this procedure in our case as it might add artifacts to data located close to the boundary. As mentioned previously, a deeper CNN model structure with a higher number of convolutional layers might also further improve predictive performance.

It must be mentioned, though, that to the best of our knowledge, all other approaches applying CNN for 3D texture predictions

transform the soil profile data to certain depth intervals prior to model training. The advantage of our approach is that it can use the soil profile data as is and does not introduce additional uncertainty to the input data by fitting depth functions which then propagates through the models. This uncertainty is currently not accounted for in any predictive mapping approach.

### 3.4 Spatial predictions

Figure 4 shows the VI of the most important predictors for the two topsoil (2D) models considering landscape patches of 5 x

5 and 10 x 10 cells. Figure 5 indicates the VI of the most important predictors for the models to predict in 3D, respectively. For the topsoil predictions elevation above sea level (DEM00), terrain surface texture with a 10 cells radius (TST10), valley depth (VAD00), vertical overland flow distance to major rivers (VOF0M), the Geomorphographic map of Germany (GMK00), the Maps of parent material (BAG00) and Stratigraphy (STR00), the Soil scapes in Germany map (BGL00), Latitude (LAT00), and Longitude (LON00) are among these predictors (Fig. 4). For the 3D predictions the list includes the same predictors and

additionally the dry matter productivity of June 2016 and 2018 (DMP16, DMP18). For the topsoil models the high importance of the categorical predictors GMK00, BAG00, STR00, and BGL00 is clearly visible with BAG00 indicating the highest VI. Depending on the considered particle size fraction, LAT00 or LON00 are of higher importance than STR00 and/or GMK00. GMK00 is only of high importance for predicting the sand fraction. Particularly, the VI values of the categorical predictors also display a high variability. This might be due to the soil profile dataset being too small to capture all categories with

sufficient profiles.



For the 3D predictions, BAG00 (Fig. 5A8 – 5F8) also is by far the most important predictor (please be aware of the different Y-axis scales). BAG00 and other predictors with high VI values, namely STR00, BGL00, LAT00, and LON00 display decreasing VI values with depth, while there is no predictor with increasing VI values. This indicates very well the reason for the increasing model uncertainty with depth. Obviously, important predictor information that could explain soil texture

variation in the subsoil is missing and cannot be captured by indicators of geographic location (LAT00, LON00), either.

The high importance of the included expert information in terms of soil and parent material vector maps was also reported by Taghizadeh-Mehrjardi et al. (2021), and Gebauer et al. (2022). However, on the contrary, it could also indicate that the CNN algorithm favors categorical dummy predictors over continuous predictors since the detection of low-level features such as edges and curves is more straightforward. This aspect is application-specific to the usage of landscape patches as predictors

for soil information, but could possibly be compared to favoring sharp edges over blurred image features in image classification tasks. It would have to be tested whether excluding vector maps encoded as dummy variables would lead to a similar model performance or even improve the results. Many studies have shown that SORPAN P predictor information could partly be replaced by topographical predictors. (Gebauer et al., 2022) demonstrated this for German topsoil texture predictions.

However, altogether the importance of SCORPAN P and SCORPAN R predictors for the prediction of the particle size

fractions in 2D and 3D is clearly visible. But the results indicate that the included predictor information is less well-suited for predicting soil texture in the subsoil.

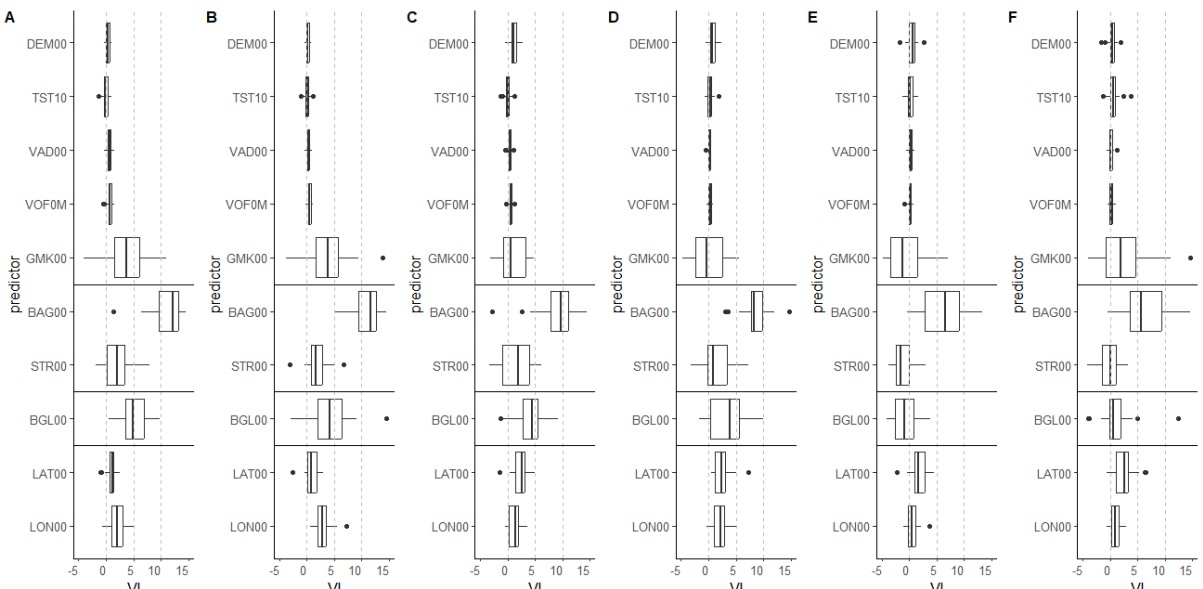

**Figure 4: Variable importance plots for the topsoil prediction model. Only predictors with a VI ≥ 0.5% in at least one of the two models for at least one of the three particle size fractions are displayed. A) Sand content [w = 5 cells], B) sand content [w = 10 cells],**
**C) silt content [w = 5 cells], D) silt content [w = 10 cells], E) clay content [w = 5 cells], and F) clay content [w = 10 cells]. Please refer to Table 1 for the predictor abbreviations.**





**Figure 5: Variable importance (VI) plots for the 3D prediction models. Only predictors with a VI ≥ 0.5% in at least one of the two models for at least one of the three particle size fractions are displayed. A) Sand content [w = 5 cells], B) sand content [w = 10 cells], C) silt content [w = 5 cells], D) silt content [w = 10 cells], E) clay content [w = 5 cells], and F) clay content [w = 10 cells]. Figure lines from top to bottom correspond to the predictor variables: 1 = DEM00, 2 = TST10, 3 = VAD00, 4 = VOF0M, 5 = GMK00, 6 = DMP16, 7 = DMP18, 8 = BAG00, 9 = STR00, 10 = BGL00, 11 = LAT00, and 12 = LON00. Please refer to Table 1 for the predictor abbreviations.**

Figure 6 displays selected horizontal slices of the 3D prediction of the particle size fractions sand, silt, and clay. Glacial deposits in the North German Lowland account for the region's high sand (Fig. 6A1- 6A4) and comparatively low silt (Fig. 6B1 – 6B4) and clay contents (Fig. 6C1 – 6C4). The predictors BAG00 and BGL00 make an important contribution to this pattern. Glaciers that advanced from Scandinavia during the three Pleistocene ice eras left behind coarse sedimentary deposits. The lower sand contents in North-East Germany correspond to the young moraine area which is intermixed with Sander and glacial valleys. The high silt and clay contents along the North Sea indicate the marsh border on the tidal coast (Liedtke and Mäusbacher, 2003). Together with the higher silt and clay contents along the Northern German major river valleys due to floodplain sediments, this coincides well with the unit of the Geomorphographic Map of Germany indicating sink areas at very





low elevation above the depth line and with very high soil moisture index. The high silt contents north of the Central German Uplands and in Southern Germany reflect Loess deposits, mostly with periglacial influence (Liedtke, 2002). In between are the escarpment and ridge landscapes with Triassic rocks and larger Loess-covered basins. The BAG00, BGL00, and GMK00 map units help to differentiate the soil texture pattern in this area. Overall, the spatial distribution indicates the strong influence of the soil parent material and topography. The topsoil pattern aligns well with that predicted by Gebauer et al. (2022).

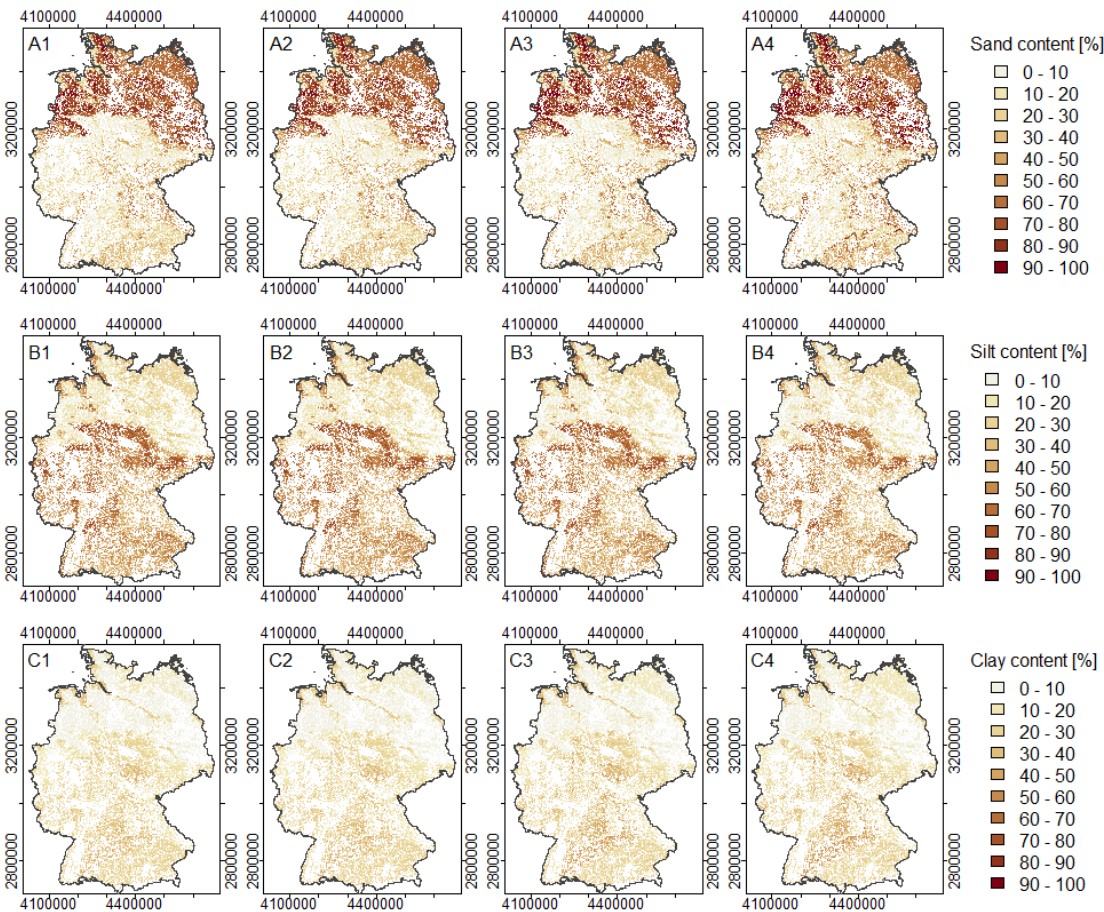

**Figure 6: Spatial predictions from the 3D model of patch size w = 10. Figure lines correspond to the three particle size fractions: A) sand content, B) silt content, and C) clay content. Figure columns display the respective value in 1, 20, 50 and 100 cm depth. Non-agricultural areas were masked out on behalf of the CORINE Land Cover Inventory 2018 (Büttner et al., 2017)**

## 4    Conclusions

Many soil functions and processes are controlled by the soil particle size distribution. The generated three-dimensional continuous data product of the particle size fractions sand, silt, and clay for the agricultural soil-landscape of Germany has a resolution of 100 m in geographic space and 1 cm in depth.  This product is an important component for predicting the effects



of agricultural management practices and their capacity to adapt to climate change and for analyzing soil functions and numerous risks.

It was demonstrated how effective the convolutional neural network algorithm is at producing such multidimensional and multivariate data products. The technique introduces numerous possibilities for predictive soil mapping: (1) Without the use of depth functions to normalize soil profile complexity in present depth layers, 3D data products can be generated straight from

soil profile data. (2) A single model can concurrently predict a variety of soil characteristics and how they interact within and between soil horizons. (3) Each soil profile's surrounding landscape can be taken into account, and numerous landscape features can be automatically retrieved. To sum up, the potential for this deep learning approach to understand and model the complex soil-landscape relation is virtually limitless. The patch-based CNN for 3D multivariate soil modeling has only data-driven limitations. To approximate the soil forming factors, access to the vast buried treasure of soil profile data and the steadily

improving availability, quality, and resolution of gridded landscape data is essential.

Overall, there is a high demand to test the required complexity and depth of convolutional neural network models to produce soil data cubes of sufficient quality without excessive use of computing capacities. The same applies to the inclusion of the landscape context surrounding each soil profile, because vicinity size, filter size, and predictor resolution likely affect one another.

**Acknowledgements**

This work is part of the SoilSpace3D-DE project.

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
