# Peer review of "Deep learning with a multi-task convolutional neural network to generate a national-scale 3D soil data product: Particle size distribution of the German agricultural soil-landscape"

_EGUsphere, 2023_

## Author Comment (AC1)

Dear Philippe, thank you for your critical reading and valuable comments. Please find our detailed replies below each of your comments:

This paper describes the predictions over Germany of particle size fractions (clay, silt and sand) using a CNN algorithm trained on around 3,000 locations with measured soil properties and classical soil covariates. There have been a lot of nation-wide applications of Digital Soil Map in the past and already some applications of CNN in Digital Soil Mapping. I therefore consider that the novelty of this paper is rather poor.

Reply: We disagree.
1. The generated three-dimensional continuous data product, which covers the particle size fractions of sand, silt, and clay in the agricultural soil-landscape of Germany, has a spatial resolution of 100 m and a depth resolution of 1 cm. It is the first data product of its kind.
2. The applied approach allows for the incorporation of soil profile data with the respective soil horizon boundaries without the need to compute target values at predefined soil depths. In contrast to this, the common depth function approach in DSM applications to transform the horizon-wise soil profile data into depth intervals introduces an additional source of uncertainty which is generally not accounted for.
3. Furthermore, the overall potential of the convolutional neural network (CNN) algorithm in generating data products that are both multidimensional and multivariate is demonstrated in this nationwide application.

We will further enhance these aspects throughout the manuscript.

However, I notice that the authors used a genetic algorithm to train the hyperparameters of their CNN, which would perhaps constitute a novelty for the application of CNN in Digital Soil Mapping if the added value of this pre-processing step should be clearly demonstrated, which is not in this present version.

Reply: Please compare lines 65-69: "*Like many other machine learning algorithms, CNNs can only develop their full potential by applying an optimization approach for hyperparameter tuning (Gebauer et al., 2022). Still, few researchers have attempted to tune the CNN hyperparameters in predictive soil mapping studies, despite recent work showing the importance (e.g. Wadoux et al., 2019; Omondiagbe et al., 2023; Taghizadeh-Mehrjardi et al., 2020).*"

Contrary to what the authors argued in the conclusion, I do not think that the efficiency of a CNN algorithm is clearly demonstrated from the results that are presented. Indeed, using CNN does not increase the performances obtained earlier on the same dataset (topsoil particle size fractions) by a more simple-to-use learning algorithm (gradient boosted tree, Gebauer et al) and obtained very poor prediction performances for particle size fractions beyond 30 cm depth (figure 3 bottom line).

Reply:
1. We understand your concern. Therefore, we critically discuss the performance of our modeling approach using the CNN algorithm in section 3.3 Predictive Performance. Please compare e.g. lines 285-290: "*Below 26 cm, the RMSE increases abruptly. At 40 cm depth, the RMSE values are 20.9, 16.5, and 11.8 mass-%. The decrease in model performance with depth has often been reported (Taghizadeh-Mehrjardi et al., 2020; Ließ, 2022; Poggio and Gimona, 2017). One reason could be the frequent changes in parent material. Considering the high percentage of profile sites with changes in parent material it is likely that the available data for SCORPAN P is causing this increase in uncertainty. The SCORPAN P proxies consist of geological maps of different scale and quality, and there may be some inconsistencies in how depth is represented. These maps may not fully capture the nuances of the underlying parent material.*"
2. We agree that there is still work that needs to be done. Please compare lines 386-389 of the Conclusions Section: "*Overall, there is a high demand to test the required complexity and depth*

*of convolutional neural network models to produce soil data cubes of sufficient quality without excessive use of computing capacities. The same applies to the inclusion of the landscape context surrounding each soil profile, because vicinity size, filter size, and predictor resolution likely affect one another.*"

3. However, we are not aware of any other machine learning algorithm applied in the context of DSM for the generation of data products that are both multidimensional and multivariate.

Accordingly, the presented manuscript indicates the high potential for CNN in pedometric modeling approaches to generate data products that are both multidimensional and multivariate, but we conclude that further investigation is required.

Furthermore, I have some additional questions and comments along the text :

L94: explain how the horizon boundaries were taken into account in the vertical sampling scheme

Reply: This reviewer's comment refers to the following statement "*Samples were taken for each of the 0-10, 10-30, 30-50, 50-70, and 70-100 cm depth increments, while taking horizon boundaries into account.*"
It means that samples were taken separately for each horizon fragment occurring within the respective depth increment. For example: If a horizon boundary occurred at 63 cm depth a sample was taken for the 50-63 cm section and another for the 63-70 cm section. We will specify this in the manuscript to add clarity.

L173: I do not understand why the authors fed their CNN with soil observations containing 100 soil layers of 1cm whereas soil particle fractions were only measured at 5 depth intervals. This uselessly overload the CNN without bringing more significant information. It consequently increases the number of parameters and hamper the convergence of the algorithm toward a satisfactory prediction.

Reply:
1. The soil particle size fractions were not only measured at depth intervals. Please compare our reply to the previous comment.
2. In soil science, horizon boundaries are usually recorded at a precision of 1 cm. To consider the horizontation in the model training the applied 1 cm resolution with depth is a fair data representation and not a useless overload.
3. We trained our CNN model to extract the best possible information from the data. Now you may argue whether the available number of soil profiles is sufficient to capture soil complexity in the agricultural soil-landscape of Germany. Please compare lines 382-385: "*… the potential for this deep learning approach to understand and model the complex soil-landscape relation is virtually limitless. The patch-based CNN for 3D multivariate soil modeling has only data-driven limitations. … access to the vast buried treasure of soil profile data and the steadily improving availability, quality, and resolution of gridded landscape data is essential.*"

L187: A thorough presentation of the importance of missing data per soil property and soil depth increment is necessary. To my experience, the numbers of missing data generally increase with depth (soils are not all 100 cm thick). This could also explain why the prediction performances collapse beyond 30 cm

Reply: Please remember these are soils under agricultural use, not soils in general. Please compare lines 186-196 for the requested details "*Table 3 indicates the size of the datasets used for model training to predict soil texture …The different sizes result from excluding predictor arrays with missing data, and/ or profiles with missing texture data in part of the profile…..The decrease from 2917 to 2740 sites …is due to missing texture data in part of the subsoil.*"

L190 : I disagree with this statement. It is quite easy to find transnational covariates as shown by the number of papers presenting continental or global applications of DSM

Reply: Soil texture predictions heavily rely on data proxies to the soil forming factor parent material. Spatially continuous representations of this factor are not often available beyond national boundaries. The same applies to expert information contained in conventional map products. We will further explain this aspect in the manuscript.

L 206 : "All predictors were recoded into dummy variables": similarity between the categorical values not taken into account?

Reply: Unfortunately, the similarity between categorical values is defined, neither for the included soilscapes map nor the parent material map. This is a common problem with conventional map products. To define this similarity is a time-consuming workload beyond the objective of this paper.

1. 238 : No data augmentation? could be easily done by rotating/mirrroring windows

Reply: The comment refers to lines 237-238: *"The VI plots exhibit boxplots of twenty-five VI values for each predictor due to the five times repeated 5-fold CV procedure (outer CV cycle)."* We decided to show the distribution of the VI values instead of the mean values.

L245-247: As a pedologist, I am very surprised to read that 44,8% of the soil observations sampled in Germany are polyphasic soils with more than one parent material. Please check this information from an experienced soil scientist.

Reply: The reviewer comment refers to lines 243-247 *"Regarding the 2740 profiles included in the dataset for 3D modeling with a 5 × 5 cell patch size, field data annotations show that 44.8% of the soil profiles have one change in parent material up to a depth of 100 cm…"*
You may easily check yourself. The complete dataset of the agricultural soil inventory (3103 profiles) has a similarly high percentage of soil profiles with at least one change in parent material within the top 100 cm.
The dataset can be accessed from Poeplau, C.; Don, A.; Flessa, H.; Heidkamp, A.; Jacobs, A.; Prietz, R. *First German Agricultural Soil Inventory–Core Dataset*; Open Agrar Repositorium: Göttingen, Germany, 2020. https://doi.org/10.3220/DATA20200203151139.
Currently, there is a bug in the reference list, which we will correct upon resubmission.

L247: A table showing the main statistical indicators of the distribution of soil properties (mean, variance, min, max etc…) and histograms would be more informative than figure 2. In particular, we need to know the variance to interpret the RMSEs that are given further

Reply: Figure 2 indicates the boxplot values of the sand silt and clay content throughout depth. We will additionally add a line to indicate the mean values.

L276: RMSE should be usefully completed by other prediction performance indicators such as R2, Model Efficiency Coefficient (MEC) or LCCC.  some scatterplots of the measured versus predicted soil properties should be added to give more insight into the behaviour of the model.

Reply: The RMSE (1) allows for the direct comparison with other soil texture data products covering Germany, which were evaluated on the same test set data, and (2) can be used to account for uncertainty propagation when using the data product. Providing additional performance indicators would provide little additional insight in this context. Scatter plots would amount to 300 individual

plots for the 3D predictions. As a compromise, we will add scatter plots with R² values for selected soil depths as supplementary material.

L284 : The bottom line curves show clearly stair-step shapes. Any interpretation of that?

Reply: Please compare Lines 285-290: "*The decrease in model performance with depth has often been reported … One reason could be the frequent changes in parent material. Considering the high percentage of profile sites with changes in parent material it is likely that the available data for SCORPAN P is causing this increase in uncertainty. The SCORPAN P proxies consist of geological maps of different scale and quality, and there may be some inconsistencies in how depth is represented. These maps may not fully capture the nuances of the underlying parent material.*"
Beyond this, we have no further explanation for the step-wise decrease in model performance with depth.

L287-291: This interpretation does not convince me. Normally, the P covariate should be more related with deep horizons than with superficial ones as the former are expected to be closer to the parent rock described in geological database. Consequently, the soil property predictions should be better for deep horizons if the limiting factor was the P covariate.

Reply: The reviewer refers to lines 285-290: "*The decrease in model performance with depth has often been reported … One reason could be the frequent changes in parent material. Considering the high percentage of profile sites with changes in parent material it is likely that the available data for SCORPAN P is causing this increase in uncertainty. The SCORPAN P proxies consist of geological maps of different scale and quality, and there may be some inconsistencies in how depth is represented. These maps may not fully capture the nuances of the underlying parent material.*"
We agree that SCORPAN P is expected to have higher explanatory power for deeper soil horizons, while the explanatory power of other soil-forming factors (R, O, C) decreases with depth. However, as a consequence of this, data proxies to SCORPAN P which may explain the parent material only to a very limited extent lead to a decrease in predictive model performance with depth. We will add this general understanding to avoid confusion.

---

## Author Comment (AC3)

Dear reviewer, thank you for your critical reading and valuable comments. Please find our detailed replies below each of your comments.

The paper uses a CNN algorithm in a DSM exercise in Germany, using a relatively large collection of soil profiles. In general, a well written manuscript but it would benefit from reducing the use of "data science" jargon and more consistent citations.

Reply: There are two sentences using the term "data science". We will replace it and review the citations.

My main concern with the manuscript is that it fails to demonstrate how their approach is more effective (as stated in the abstract and conclusions). They only provide results for two CNN variations without comparing it with conventional DSM models (without spatial context), and they obtain inferior performance compared to previous studies using the same dataset. In addition to that, they use 1 cm slices instead of using a depth function stating that it is better but without showing any results to support it.

Reply:
**CNN Approach:**
1. We are not aware of any other machine learning algorithm applied in the context of DSM for the generation of data products that are both multidimensional and multivariate. The sentence in the abstract reads "*The effectiveness of the convolutional neural network (CNN) algorithm in producing multidimensional, multivariate data products is demonstrated*"
2. We understand your concern with regard to model performance. Therefore, we critically discuss the performance of our modeling approach using the CNN algorithm in section 3.3 Predictive Performance.
3. We agree that there is still work that needs to be done. Please compare lines 386-389 of the Conclusions Section: "*Overall, there is a high demand to test the required complexity and depth of convolutional neural network models to produce soil data cubes of sufficient quality without excessive use of computing capacities. The same applies to the inclusion of the landscape context surrounding each soil profile, because vicinity size, filter size, and predictor resolution likely affect one another.*"

Accordingly, the presented manuscript indicates the high potential for CNNs in pedometric modeling approaches to generate data products that are both multidimensional and multivariate, but we conclude that further investigation is required.

**1 cm slices:**
Samples were taken as bulk samples mixed from several samples throughout a soil horizon, a common if not the standard procedure in soil surveys. Horizons in soil science are mostly defined by noticeable changes from one horizon to another in terms of color and/ or texture. In line with this, the soil texture data of the agricultural soil inventory does not correspond to a specific depth but a depth interval. Accordingly, it is reasonable to assign to each centimeter within each depth interval the same value.

In contrast, training a depth function usually requires assigning the bulk sample value to a specific soil depth. It then models gradual changes toward the mean values of the above and below-lying horizons and, thereby, somehow counteracts the rationale behind the horizon-wise sampling procedure.

**Specific comments**

- The abstract needs more work. It reads like the summaries for non-experts that some journals require.

Reply: We will add further details to the abstract.

- L56: Behrens et al. (2018) did not use a CNN.

Reply: Thank you. We will replace it with other references.

- L61: You are talking about CNNs applied in the context of soil mapping but, again, some of the references are not related to that (Behrens et al. (2010) and Behrens et al. (2014)). Considering that the list is not very long, you are missing some references.

Reply: Thank you. We will revise the list of references.

- L68: I am not sure that this is true. In most of publications, I have seen have some hyperparameter optimisation such as grid or random search.

Reply: The reviewer refers to lines 65-86: "*Like many other machine learning algorithms, CNNs can only develop their full potential by applying an optimization approach for hyperparameter tuning (Gebauer et al., 2022). Still, few researchers have attempted to tune the CNN hyperparameters in predictive soil mapping studies, despite recent work showing the importance (e.g. Wadoux et al., 2019; Omondiagbe et al., 2023; Taghizadeh-Mehrjardi et al., 2020).*"
We disagree. Grid or random search are parameter tuning techniques but do not involve an optimization approach.

- L73: is the 3D model often worse than the 2D? any reference?

Reply: Please refer to lines 285-287 (discussion section): "*The decrease in model performance with depth has often been reported (Taghizadeh-Mehrjardi et al., 2020; Ließ, 2022; Poggio and Gimona, 2017).*"

We'd rather refrain from additionally adding it to the last paragraph of the introduction section. Its last sentence merely gives the explanation of why we include the topsoil predictions as a benchmark. Lines 69-74: "*We will demonstrate an approach for implementing multivariate regression to generate a national-scale data product of the 3D spatial variation of the particle size distribution for the agricultural soil-landscape of Germany. It will be obtained with a single model employing a patch-wise multi-target CNN to predict three particle size fractions, sand, silt, and clay simultaneously at high vertical resolution until 100 cm depth. Genetic algorithm optimization will be applied for hyperparameter tuning. A CNN model to generate a topsoil data product (2D) is included to provide a benchmark since often the more complex 3D model training results in a lower performance.*"

- L125: You used coordinates as covariates, hoping to represent spatial patterns that other covariates do not capture. What kind of patterns would that be?

Reply: Pedogenesis in Germany has been ongoing for more than 10,000 years while the data we use to approximate the soil-forming factors relates to the last decades only. So there will always be some gap that we seek to cover by including spatial position. We will add this explanation to the manuscript.

- L135-146: I understand that you are trying to summarise a lot of concepts in a single paragraph but it does not read well and it is very inaccurate. E.g. the description of the learning rate is very simplistic. A high value not always speeds up the learning process and a low value not always ensures that the network succeeds in learning the predictor-response relation. In general, I understand what you mean because I have worked with CNNs but another reader will not get any value from this.

Reply: We will adapt the text section to improve understandability.

- L150-162: A lot of "data science" jargon here. Also, you are constantly mixing CNNs, CNNs applied to spatial modelling and the specific CNN architecture that you use. Please, do not mix them all in one paragraph. For instance, towards the end you mention that "the output is flattened before it enters a sequence of dense layers". That is specifically for your CNN but the text reads as if it is true for all CNNs.

Reply: Thank you. We will better separate the structure of our CNN models from the general explanation. We do not understand what you mean by "data science" jargon, though. Could you provide examples?

- L167: "CNNs cannot handle this type of input". I am OK with your pragmatic approach of limiting the window size to avoid missing data but CNNs can handle missing data. I assume that you are specifically talking of missing data represented by the float "NA".

Reply: We would be glad to learn which CNN implementation could handle missing data without replacing them previously. Missing data in R are represented by NA.

- Section 2.6: A lot of "new" genetic algorithm jargon. Islands? Migration? I do not think GA is common enough to skip those concepts. The reader would benefit with a brief introduction of the algorithm that you used.

Reply: Thank you. We will adapt this section.

- Section 2.7: No reference to the method? It sounds like a ad hoc implementation of Shapley values but you do not specify any of the details. Number of permutations? All the predictor simultaneously? Please add more details.

Reply: The reviewer refers to Lines 232-238: *"Each predictor's importance was determined by permuting the predictor in the test set prior to model application. Any predictorresponse association relating to that predictor was thus deleted. The resulting relative increase in the predictive RMSE was then assigned to the respective predictor as variable importance (VI). This VI estimation was performed for each of the three particle size fractions as well as for all depth slices (3D prediction). The values from five permutations were averaged. The VI values for the dummy variables were generated by aggregating each categorical predictor. The VI plots exhibit boxplots of twenty-five VI values for each predictor due to the five times repeated 5-fold CV procedure (outer CV cycle)."*

We will add a reference for permutation-based variable importance calculations. The number of permutations is given in line 236.

- Table 3: I have seen kernels with even number of pixels (e.g. 2x2) in a couple of DSM publications and still have not seen a justification of why an asymmetrical convolution would be desirable (they introduce aliasing errors). That is why in signal processing, kernel operations such as convolutions are

often preferred to be symmetrical (e.g. 3x3). You need to be careful when defining the search space for your hyperparameter tuning.

Reply: The search space for hyperparameter tuning only considers symmetrical kernels. Please refer to Table 2, parameter P2: kernel size in both dimensions.

- Section 3.2: You are missing information about the convergence of the GA optimisation. Also, did you get any insights from this process? You have a population of 500 individuals, and assuming that you ran it for at least 20 generations, you trained 10,000 models. In my experience, that is much longer than a well defined grid search. For instance, a dropout rate of 0.1019826 (from your 3D, 5 cells model) is not different from a dropout rate of 0.1.

Reply:
Please refer to Lines 282-230: *"The total number of iterations was set to 200, and the number of consecutive generations without an improvement in the best fitness value before the GA was terminated was set to 20."*

To test only a very limited set of selected tuning parameter combinations (grid search) may result in good model performance or it may not. The trouble is how do you select the right values and combinations while continuous parameters are involved? In these cases, an optimization approach pays off. Otherwise, you might risk missing the optimal tuning parameter set.

- L274. I would not call that "uncertainty of the model predictions".

Reply: The sentence will be adapted.

- L306. Did you try it and observed artifacts? It is quite common to assign -1 or other values to missing data.

Reply: The convolutional layers in CNNs are looking for spatial patterns. For the area outside Germany we would assume a single value while replacing missing data by -1, and highlight the national border. Consequently, we might risk extracting the national border as a feature for e.g. parent material.

- L309-312: You mentioned that your method does not introduce additional uncertainty (compared to standard intervals methods such as equal area spline) but that is not necessarily true. Since you subdivided into 1cm slices, I assume you have the same value for each slice within the original layer (e.g. 10 slices with the same clay content within a 0-10cm layer). That procedure is also a depth function but defined by you instead of fitted to data. If you could show that this method is actually better than the traditional DSM approach, it would be a valuable contribution.

Reply:
Samples were taken as bulk samples mixed from several samples throughout a soil horizon, a common if not the standard procedure in soil surveys. Horizons in soil science are mostly defined by noticeable changes from one horizon to another in terms of color and/ or texture. The available soil texture data does not correspond to a specific depth but a depth interval. Accordingly, it is reasonable to assign to each centimeter within each depth interval the same value.

Training a depth function usually requires assigning the bulk sample value to a specific soil depth. It then models gradual changes toward the mean values of the above and below-lying horizons and, thereby, somehow counteracts the rationale behind the horizon-wise sampling procedure.

- Section 3.4: Interesting that the model mostly uses categorical covariates. How many of the 119 predictors are "dummy" classes? Did you normalise/standardised the contiguous covariates?

Reply: 37 of the 119 predictors are dummy classes originating from the categorical covariates. All contiguous covariates were scaled to the range 0-1. Please refer to lines 207-208.